# Development and validation of an LC-MS/MS method for determination of hydroxychloroquine, its two metabolites, and azithromycin in EDTA-treated human plasma

**Vong Sok, Florence Marzan, David Gingrich, Francesca Aweeka, Liusheng Huang** *

Drug Research Unit, Department of Clinical Pharmacy, School of Pharmacy, University of California at San Francisco, San Francisco, California, United States of America

* Liusheng.huang@ucsf.edu

**Data Availability Statement:** All relevant data are within the manuscript and its Supporting Information files.

## Abstract

### Background

Hydroxychloroquine (HCQ) and azithromycin (AZM) are antimalarial drugs recently reported to be active against severe acute respiratory syndrome coronavirus- 2 (SARS-CoV-2), which is causing the global COVID-19 pandemic. In an emergency response to the pandemic, we aimed to develop a quantitation method for HCQ, its metabolites desethylhydroxychloroquine (DHCQ) and bisdesethylchloroquine (BDCQ), and AZM in human plasma.

### Methods

Liquid chromatography tandem mass spectrometry was used to develop the method. Samples (20 µL) are extracted by solid-phase extraction and injected onto the LC-MS/MS system equipped with a PFP column (2.0 × 50 mm, 3 µm). ESI$^+$ and MRM are used for detection. Ion pairs $m/z$ 336.1→247.1 for HCQ, 308.1→179.1 for DHCQ, 264.1→179.1 for BDCQ, and 749.6→591.6 for AZM are selected for quantification. The ion pairs $m/z$ 342.1→253.1, 314.1→181.1, 270.1→181.1, and 754.6→596.6 are selected for the corresponding deuterated internal standards (IS) HCQ-d$_4$, DHCQ-d$_4$, BDCQ-d$_4$, and AZM-d$_5$. The less abundant IS ions from $^{37}$Cl were used to overcome the interference from the analytes.

### Results

Under optimized conditions, retention times are 0.78 min for BDCQ, 0.79 min for DHCQ, 0.92 min for HCQ and 1.87 min for AZM. Total run time is 3.5 min per sample. The calibration ranges are 2–1000 ng/mL for HCQ and AZM, 1–500 ng/mL for DHCQ and 0.5–250 ng/mL for BDCQ; samples above the range are validated for up to 10-fold dilution. Recoveries of the method ranged from 88.9–94.4% for HCQ, 88.6–92.9% for DHCQ, 88.7–90.9% for BDCQ, and 98.6%-102% for AZM. The IS normalized matrix effect were within (100±10) % for all 4 analytes. Blood samples are stable for at least 6 hr at room temperature. Plasma

**Funding:** The APC was funded by UCSF library. No additional external funding was received for this study.

**Competing interests:** The authors have declared that no competing interests exist.

samples are stable for at least 66 hr at room temperature, 38 days at -70°C, and 4 freeze-thaw cycles.

## Conclusions

An LC-MS/MS method for simultaneous quantitation of HCQ, DHCQ, BDCQ, and AZM in human plasma was developed and validated for clinical studies requiring fast turnaround time and small samples volume.

## 1. Introduction

The new coronavirus disease 2019 (COVID-19), caused by severe acute respiratory syndrome coronavirus 2 (SARS-CoV-2), has evolved into a world pandemic since the first 4 cases were reported in December 29, 2019, in Wuhan, China [1]. As of August 25, 2020, there are 23.5 million COVID-19 cases worldwide with 810,492 deaths [2], among which 5.77 million cases with178,129 deaths are in USA [3]. There is an urgent need for effective drugs to treat COVID-19. Initial studies found hydroxychloroquine (HCQ) active against SARS-CoV-2 [4] and potentially useful for the treatment of COVID-19 illness clinically [5] and azithromycin (AZM) was evaluated in combination with HCQ to treat COVID-19 [5]. However recent clinical trials, lacking critical pharmacology evaluations to inform optimal dosing and requiring drug quantitation methods, have reported substantial toxicities and contradicted results. Some studies reported benefits [5–7] while others reported no benefits [8–10].

HCQ, primarily used previously as an antimalarial drug, has also been used for autoimmune diseases such as rheumatoid arthritis for several decades [11, 12]. HCQ is 50% bound to plasma proteins, absorbed completely and rapidly(70–80% in the gastrointestinal tract) [13] and is characterized by a long half-life(up to 40 days). Its peak concentration($C_{max}$) in the context of multiple dosing may reach up to 1000 ng/mL [14, 15]. Hepatically, HCQ is metabolized by cytochrome p450 (CYP) 2D6 to desethyl-chloroquine (DCQ) and desethyl-hydroxychloroquine (DHCQ)—both of which exhibiti activity for rheumatoid arthritis [16]; as well as bis-desethyl-hydroxychloroquine (BDCQ), a metabolite implicated in HCQ toxicity [17, 18]. At steady state, DHCQ in blood reaches approximately the same concentration as HCQ while BDCQ exhibits ~1/10 of HCQ concentrations [18].

AZM is a $2^{nd}$ generation macrolide antibacterial that inhibits bacterial protein synthesis [19]. It also exhibits moderate activity against malaria and is used in combination with chloroquine for malaria chemoprevention [20] and treatment [21]. AZM $C_{max}$ has been reported to be ~400 ng/mL following a 500 mg single dose [22], with higher $C_{max}$ expected following multiple doses due to its long half-life of ~70 hrs. AZM is characterized by both low oral bioavailability(17–37%) and low plasma protein binding (~30%) [19]. It accumulates in tissues and blood leukocytes.

As our group is a leading pharmacology laboratory for HIV and malaria and as part of the wide-spread international emergency response to the outbreak of COVID-19, our laboratory rapidly developed, validated and received approval from a NIH Division of AIDS (DAIDS) supported quality assurance program, for a liquid chromatography tandem mass spectrometry (LC-MS/MS) method. This method was developed to support clinical trials and to assess the pharmacokinetics (PK) and pharmacodynamics (PD) of HCQ, DHCQ, BDCQ and AZM. LC-MS/MS is the preferred technique for drug analysis due to its high sensitivity and selectivity. While LC-MS/MS methods to quantitate HCQ [13, 23, 24] and its metabolites [17, 18, 25]

have been reported, they are mainly for analyzing whole blood samples. A number of LC-MS/MS methods have also been published to measure AZM in human plasma [26, 27]. To the best of our knowledge, this is the first method for the simultaneous quantitation of HCQ, its metabolites and AZM in human plasma. Previous studies reported plasma/serum HCQ ranges from 1.0 to 2440 ng/mL with the majority of samples being between 50.0–1700 ng/mL [24] while AZM $C_{max}$ in plasma ranges from 200 ng/mL to 1500 ng/mL depending on the dosage [19]. Therefore, the assay calibration curve ranges were tailored for 2–1000 ng/mL for both AZM and HCQ, 1–500 ng/mL for DHCQ, and 0.5–250 ng/mL for BDCQ. This assay requires only 20 μL plasma sample volume.

## 2. Materials and methods

### 2.1. Materials

Azithromycin and hydroxychloroquine (Fig 1) were USP reference standards purchased from Sigma-Aldrich. Desethyl-hydroxychloroquine, bisdesethyl-chloroquine, the internal standards azithromycin-d5, hydroxychloroquine-d4, Desethyl-hydroxychloroquine-d4, and bisdesethyl-chloroquine-d4 were purchased from Toronto Research Chemicals. Trifluoroacetic acid, acetonitrile, methanol and water were purchased from Thermo-fisher (Optima LC/MS grade). Blank human plasma and blood ($K_2$ or $K_3$ EDTA added as anticoagulants) was obtained from Biological Specialty Co (Comar, PA, USA).

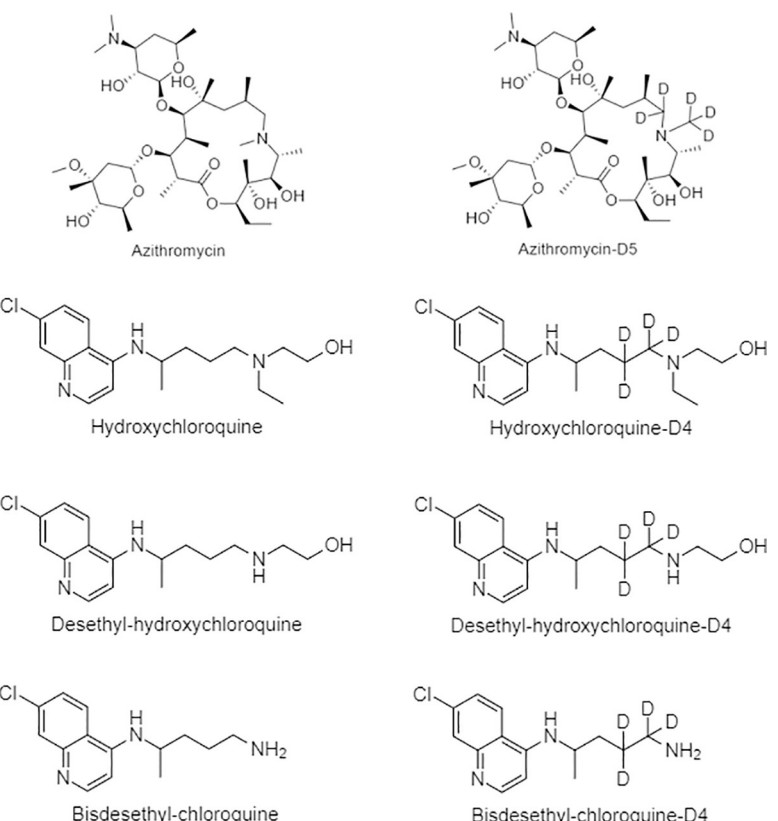

**Fig 1. Chemical structures of azithromycin, hydroxychloroquine, desethylhydroxychoroquine, bisdesethylchloroquine and the internal standards.**

## 2.2. Instrumentation

Sciex API5000 tandem mass spectrometer was coupled with a Shimadzu Prominence 20ADXR LC pumps and SIL-20ACXR autosampler. The LC column was Pursuit pentafluoro-phenyl (PFP) (50×2.0 mm, 3μm) fitted with a guard column (10×2.0 mm, 3μm) (Agilent Tech., Santa Clara, CA, USA) and eluted with water (A) and acetonitrile (B) both containing 0.05% trifluoroacetic acid (TFA) at a flow rate of 0.5 mL/min in a gradient mode: 20% solvent B (0–0.2 min), 20 to 50% B (0.2–1.5 min), 50–90% B (1.5–1.6 min), 90 to 100% B (1.6–2.0 min), 100% B (2.0–2.5 min), 100 to 20% B (2.5–2.6 min), and 20% B (2.6–3.5 min). Electro Spray ionization in positive mode (ESI$^+$) was used as the ion source with multiple reaction monitoring (MRM) of $m/z$ 749.6→591.6 for AZM, $m/z$ 336.1→247.1 for HCQ, $m/z$ 308.1→179.1 for DHCQ, and $m/z$ 264.1→179.1 for BDCQ. The ion pairs for their corresponding internal standards (IS) were m/z 754.6→596.6 for AZM-d$_5$,342.1→253.1 for HCQ-d$_4$, 314.1→181.1 for DHCQ-d$_4$, and 270.1→181.1 for BDCQ-d$_4$. Samples were diverted into MS source between 0.5–2.3 min.

## 2.3. Preparation of stock, calibration standards, and quality control samples

HCQ stock solution was prepared in water. DHCQ and BDCQ stock solutions were prepared in methanol. AZM stock solution and all working solutions were prepared in methanol-water (1:1, v/v). Nine combined calibration standard samples at concentrations of 2, 5, 10, 20, 50, 100, 200, 500 and 1000 ng/mL for AZM and HCQ; 1, 2.5, 5, 10, 25, 50, 100, 250, and 500 ng/mL for DHCQ; 0.5, 1.25, 2.5, 5, 12.5, 25, 50, 125, 250 ng/mL for BDCQ were prepared in blank EDTA-treated human plasma by serial dilution from a combined working solution of AZM/HCQ/DHCQ/BDCQ (40/40/20/10 μg/mL). Quality control (QC) samples QC-L (6.00/6.00/3.00/1.50 ng/mL for AZM/HCQ/DHCQ/BDCQ), QC-M(60.0/60.0/30.0/15.0 ng/mL for AZM/HCQ/DHCQ/BDCQ), and QC-H (800/800/400/200 ng/mL for AZM/HCQ/DHCQ/BDCQ) were prepared in blank plasma from different stock solutions or the same verified stock solutions as those used for calibrators. All solutions and plasma samples were stored at -70˚C before use.

## 2.4. Sample preparation

Hydrophilic lipophilic balance (HLB) solid phase extraction micro-elution 96-well plates were preconditioned with 200μL MeOH and 200μL water sequentially. Plasma samples (20 μL) were added in the wells containing 40 μL 0.1N NaOH, Then 20 μL combined IS solution(100 ng/mL AZM-d$_5$, 40 ng/mL HCQ-d$_4$, 20 ng/mL DHCQ-d$_4$, and 20 ng/mL BDCQ-d$_4$ in 50% MeOH) was added and mixed briefly. The wells were washed with 200 μL water, followed by 200 μL 10% MeOH under vacuum, and eluted with 25 μL MeOH containing 0.5%FA under gradually increased vacuum. Elution was repeated for the second time with another 25 μL eluent. To the collection plate, 150 μL water was added with a 12-channel pipette to constitute a final volume of 200 μL and mixed by pipetting up and down three times. Sample was then injected into the LC-MS/MS at 1 μL.

## 2.5. Validation

The method was validated in accordance with guidelines outlined by both the NIH-sponsored Clinical Pharmacology Quality Assurance Program (CPQA) [28] and the FDA [29]. A full validation includes precision and accuracy, dilution integrity, selectivity, matrix effect and recovery, and stability. Dilution integrity was evaluated by diluting the extra-high QC sample (4000/

4000/2000/1000 ng/mL AZM/HCQ/DHCQ/BDCQ) by 10-fold with blank plasma. Stability in plasma was evaluated at -70˚C, room temperature and after 4 freeze-thaw cycles by comparing the treated samples with untreated samples in plasticmicrocentrifuge tubes. To evaluate auto-sampler stability, the processed low and high QC samples were first tested on the same day of processing (as control) and 3 days after having been in the autosampler. Solution stability was evaluated by diluting the solutions to within the calibration range with methanol-water (1:1, v/v). To test stability in blood, blank blood was spiked with analytes at high QC levels and mixed gently on a rotator for 5 min before centrifuging at 2000 g for 10 min to obtain plasma, which was analyzed along with freshly spiked calibrators and QCs. The remaining blood was rotated briefly and left on benchtop. A series of plasma samples was then prepared from this blood at 1hr, 2hr, 4hr, and 6hr. Each timed sample was processed and analyzed immediately following preparation. All measurements were performed in triplicates at the minimum. Selectivity was evaluated with 6 different lots of human plasma with $K_3$EDTA as the anticoagulant.

Matrix effect (ME), recovery (RE) and process efficiency (PE) were evaluated with three sets of samples: Set 1 samples were prepared by spiking 20 μL analytes in 50% MeOH solution at QC-L, QC-M, QC-H concentrations and 20 μL IS solution (100/40/20/20ng/mL AZM-$d_5$, HCQ-$d_4$, DHCQ-$d_4$, and BDCQ-$d_4$) into 160 μL 50% MeOH and analyzed in triplicates. Set 2 samples were spiked at the same concentration as Set 1 in extracted solutions from blank plasma in triplicate. Set 3 samples were prepared by spiking analytes in blank plasma with final concentrations of 6/6/3/1.5 ng/mL (QC-L), 60/60/30/15 ng/mL (QC-M), and 800/800/400/200 ng/mL (QC-H) for AZM/HCQ/DHCQ/BDCQ. These plasma samples were then processed in triplicate using protocols as described above.

To test impact of hemolysis on plasma sample analysis, a 1 mL aliquot of whole blood underwent 3 freeze-thaw cycles to lyse the blood cells. Fifty microliter of the treated blood was spiked into 950 μL plasma in triplicates to give 2–3% hemolyzed plasma, which were spiked with AZM/HCQ/DHCQ/BDCQ at QC-L and QC-H concentrations. The prepared QC-L and QC-H were processed and analyzed along with freshly spiked calibrators and QCs.

Clinical samples are likely to be collected in tubes with $K_2$EDTA instead of $K_3$EDTA as the anticoagulant. To test the impact on quantification, QC-L and QC-H were prepared in two lots of $K_2$EDTA plasma and one lot of $K_3$EDTA plasma as the control. Triplicates of these samples were processed and analyzed along with a set of calibrators.

## 3. Results and discussion

### 3.1. Method development

**3.1.1. LC-MS/MS optimization.** The LC-MS/MS system was optimized in both APCI⁺ and ESI⁺ modes. Initially APCI⁺ was chosen for its less matrix effect and lower baseline signal. However, APCI⁺ limited linearity to a range narrower than that desired (S1 Fig). ESI⁺ ion source was finally chosen for this assay. The optimized MS/MS parameters are shown in Table 1. LC separation parameters were adopted from a previous assay we had developed for piperaquine [30]. It was further discovered that mobile phases 0.05% TFA in water and 0.05% TFA in acetonitrile gave similar peak shapes and retention to those obtained using 20 mM NH₄FA 0.14% TFA in water and 0.1% TFA in acetonitrile. Therefore, the former solvent combination was used due to its simplicity. The retention times are 0.78 min for BDCQ, 0.79 min for DHCQ, 0.92 min for HCQ and 1.87 min for AZM. Total run time is 3.5 min per sample.

**3.1.2. IS selection.** The ideal IS for LC-MS/MS methods are stable isotopically-labelled analytes. In this method, we were able to obtain deuterated IS. To avoid cross talks from analytes, the less abundant ion pairs from the ³⁷Cl isotope (i.e. the most abundant ion plus 2) were

**Table 1. Optimized MS/MS parameters.**

| Source parameters | TEM, ˚C | IS, v | CAD, psi | CUR, psi | Gas1, psi | Gas2, psi |
|---|---|---|---|---|---|---|
| | 500 | 1250 | 12 | 25 | 50 | 40 |
| Compound parameters | DP, v | EP, v | CE, v | CXP, V | Dwell time, ms | |
| 749.6/591.6 (AZM) | 50 | 10 | 40 | 39 | 50 | |
| 754.6/596.6 (AZM-d5) | 50 | 10 | 40 | 39 | 50 | |
| 336.1/247.1 (HCQ) | 50 | 10 | 29 | 16 | 50 | |
| 342.1/253.1 (HCQ-d4) | 50 | 10 | 29 | 16 | 50 | |
| 308.1/179.1 (DHCQ) | 50 | 10 | 31 | 16 | 50 | |
| 314.1/181.1 (DHCQ-d4) | 50 | 10 | 31 | 16 | 50 | |
| 264.1/179.1 (BDCQ) | 50 | 10 | 30 | 16 | 50 | |
| 270.1/181.1 (BDCQ-d4) | 50 | 10 | 30 | 16 | 50 | |

TEM, source temperature; IS, ionspray voltage; CUR, curtain gas, Gas1, nebulizer gas; gas2, auxiliary gas; CAD, collision-activated dissociation; DP, declustering potential; EP, entrance potential; CE, collision energy; CXP, collision cell exit potential.

selected for HCQ-d$_4$, DHCQ-d$_4$ and BDCQ-d4. For AZM-d$_5$, a higher concentration was used to avoid cross talk from AZM especially at ULOQ level.

Selection of the appropriate concentrations for stable isotopically labelled internal standards is based on both the number of stable isotopic atoms in the IS and the mass abundance of analytes at the IS mass levels. The IS signal originated from analyte at ULOQ should not be more than 5%. The naturally occurring isotopic masses of the 4 analytes are calculated using an online calculator and shown in Table 2 [31]. The percentage is based on the most abundant mass(EM).

To avoid interference of cross talk signal from the analyte, the formula to calculate the minimum concentration of stable-isotope labelled internal standard is as follows [32]:

$$C_{IS,min} = m\% \times \frac{ULOQ}{5\%} \tag{1}$$

Where m is the cross-signal percentage from analyte to IS.

To avoid interference of cross talk signal from the IS to the analyte, the formula to calculate the maximum IS concentration is as follows:

$$C_{IS,max} = 20\% \times \frac{LLOQ}{n\%} \tag{2}$$

Where n is the cross-signal percentage from IS to analyte.

**Table 2. Natural abundances of isotopic mass of analytes and minimal IS concentration.**

| | AZM | HCQ | DHCQ | BDCQ |
|---|---|---|---|---|
| Exact Mass (EM) | 748.51 | 335.18 | 307.15 | 263.12 |
| EM+4 | | 0.740% | 0.601% | 0.408% |
| EM+5 | 0.052% | | | |
| EM+6 | | 0.003% | 0.002% | 0.001% |
| $C_{IS,min}$, ng/mL | 10.4 | 148 | 120 | 81.6 |
| *$C_{IS,min}$, ng/mL | | 0.6 | 0.4 | 0.2 |

*, [37]Cl isotope used for HCQ-d$_4$, DHCQ-d$_4$, and BDCQ-d$_4$.

According to the equations, IS concentrations higher than 100 ng/mL are needed for HCQ, DHCQ and BDCQ if the most abundant IS ions (EM+4) are selected. However, in order to prevent the spiking of signals at LLOQ levels by highly concentrated IS, we chose the less abundant $^{37}Cl$ isotope signal (EM+6) for HCQ-$d_4$, DHCQ-$d_4$, and BDCQ-$d_4$. The final IS concentrations were 100ng/mL AZM-$d_5$, 40 ng/mL HCQ-$d_4$, 20 ng/mL DHCQ-$d_4$, and 20 ng/mL BDCQ-$d_4$.

**3.1.3. Sample preparation.** Solid phase extraction (SPE) was used in this assay as it yielded a cleaner extract than protein precipitation method did. HLB microelution plate was selected in consideration ofthe small sample volumes. While the PRiME HLB plate is more user-friendly for its omission of well preconditioning step, samples loaded into its wells would gradually drain away, leaving little time for the user to homogenize sample with IS and reagents. As a result, we chose the traditional HLB plate as it allows plasma samples to be homogenized with IS and reagents in the wells prior to passing the mixture onto the stationary phase. Methods in literature utilized liquid-liquid extraction with alkalized organic solvents for AZM extraction [26, 33] and protein precipitation with acidified organic solvents for extraction of HCQ and its metabolites [25]. Considering these analytes are weak bases, we alkalized plasma samples with NaOH to help retain analytes on SPE absorbent and acidified elution solvent to help elute analytes. The combined effect improved recovery greatly.

## 3.2. Method validation

**3.2.1. Calibration curves.** Nine combined calibration standards, prepared in $K_3EDTA$ human plasma, consisting of 2, 5, 10, 20, 50, 100, 200, 500 and 1000 ng/mL for AZM and HCQ; 1, 2.5, 5, 10, 25, 50, 100, 250, and 500 ng/mL for DHCQ; 0.5, 1.25, 2.5, 5, 12.5, 25, 50, 125, 250 ng/mL for BDCQ were used to establish the calibration curves. At the lower limit of quantitation (LLOQ) (2/2/1/0.5 ng/mL AZM/HCQ/DHCQ/BDCQ) the S/N ratios were 47 for AZM, 75 for HCQ, 31 for DHCQ, and 18 for BDCQ. The calibration curves were constructed using concentration vs. peak area ratio fitted with least square linear regression weighted by 1/x for HCQ, DHCQ, and BDCQ while quadratic fitting weighted by $1/x^2$ was needed for AZM for better accuracy at lower concentrations. The relative error(%RE) sum, defined as the sum of absolute %RE values, was used to evaluate the goodness of fit when using different weighting factors for calibration curve [34]. The %RE sum of calibrators for the 4 intra/inter-day A&P runs were the lowest for quadratic $1/x^2$ weighted curve compared to those of linear regression. To compare regression models with different parameters, an effective way is to use Akaikes information criterion (AIC) [35]. This criterion not only takes into account the sum of squares of relative errors (SSR), it also includes a term proportional to the number of parameters used. AIC is calculated via the formula:

$$AIC = n \times \ln(SSR) + 2M \tag{3}$$

Where n is the number of calibrators and M is the number of parameters. The model producing the smallest AIC is preferred. Quadratic regression weighted by $1/x^2$ gave the least AIC values (Table 3).

The correlation coefficient(r) was typically >0.995. Representative chromatograms for blank plasma, the lower limit of quantification (LLOQ) and its IS, and double blank plasma injected after ULOQ and its IS are shown in Fig 2.

Fig 2 (red dash line) demonstrates that there are no significant analyte signal and cross-talk from the IS (<20% of LLOQ) for all analytes. Compared to calibrator#1 [LLOQ, where peak area = 2530 (AZM), 29900 (HCQ), 6240 (DHCQ), and 8490 (BDCQ)], the peak areas in blank samples are minimal: 94.3 (3.7%) for AZM, 839 (2.8%) for HCQ, 364 (5.8%) for DHCQ, and 775 (9.1%) for BDCQ.

**Table 3. Comparison of regression models.**

| | %RE sum | | | | AIC | | | |
|---|---|---|---|---|---|---|---|---|
| Run ID | 2 | 8 | 12 | 17 | 2 | 8 | 12 | 17 |
| Linear, 1/x weighted | 46 | 96 | 82 | 31 | 60 | 72 | 67 | 51 |
| Linear, $1/x^2$ weighted | 32 | 78 | 41 | 25 | 49 | 64 | 56 | 45 |
| Quadratic, 1/x weighted | 22 | 82 | 36 | 20 | 48 | 69 | 54 | 45 |
| Quadratic, $1/x^2$ weighted | 17 | 60 | 32 | 22 | 43 | 43 | 50 | 45 |

Fig 2 (grey line) displays each analyte's and IS's carryover signals relative to their signal intensities in LLOQ. The peak areas at the retention times of analytes in double blank samples injected after ULOQ for AZM, HCQ, DHCQ, and BDCQ are 217 (8.6%), 3000 (10%), 1050 (16.8%), and 1170 (13.9%), respectively, all within 20% of LLOQ signal (Left panel), and no peaks were found at the retention time of ISs in the chromatograms of double blank samples after ULOQ for AZM-$d_5$, DHCQ-$d_4$, BDCQ-$d_4$ (right panel). The peak for HCQ-$d_4$ in the double blank following ULOQ was not significant: peak area = 1250, representing 1.5% IS signal (83800). The results suggest carryover for all analytes and ISs are not significant. Furthermore, the IS signals from ULOQ of all analytes are no more than the IS signals from the LLOQ, suggesting the cross talk from analytes are negligible (S2 Fig).

**3.2.2. Intra-/inter-day precision and accuracy.** Precision is the degree of reproducibility; it characterizes the degree of agreement among a series of individual measurements. Precision is calculated as the coefficient of variation (%CV). Accuracy is the degree of correctness and is expressed as the percent deviation from the true concentration value. Precision and accuracy (P&A) of method should be no more than 15% except for the lower limit of quantitation (LLOQ), where ≤20% is acceptable. For inter-assay precision and accuracy, at least 3 runs with at least 5 replicates of LLOQ, low, medium, and high concentration validation samples in each run should be performed. These samples are designated as LLOQ, QC-L, QC-M, and QC-H with AZM/HCQ/DHCQ/BDCQ concentrations of 2/2/1/0.5 ng/mL, 6/6/3/1.5 ng/mL, 60/60/30/15 ng/mL, and 800/800/400/200 ng/mL, respectively.

During the validation, we found the IS solutions were not stable in glass vial especially for hydroxychloroquine and its metabolites. This resulted in unacceptable data in two inter-day P&A runs. Further stability test between storage in glass and in plastic Eppendorf tubes revealed all ISs lost signals in a few hours in glass vial but remained stable in Eppendorf tube for at least overnight. AZM-$d_5$ was reduced by ~20% depending on the container size. HCQ-$d_4$, DHCQ-$d_4$ and BDCQ-$d_4$ in 50% MeOH were reduced much more significantly (over 50%) due to adsorption on glass surface (S1 Table).

Among the 4 runs of intra-/inter-day P&A in this report, the 1st run was performed with freshly prepared calibrators and QCs from separately weighed stocks. In summary, intra- day P&A meet the criteria except for one of four runs at the LLOQ for AZM (dev +25%) using 1/x weighted linear regression calibration curve. When we reanalyzed the intra-/inter-day P&A data for AZM using quadratic regression of $1/x^2$ weighted calibration curve, all 4 runs passed the acceptance criteria.

The inter-assay precisions(%CV) of the method at low, medium, and high concentrations are 7.5%, 6.9%, 4.9% for AZM; 7.3%, 5.3%, 6.2% for HCQ; 7.5%, 6.3%, 5.1% for DHCQ; 13%, 7.4%, 8.0% for BDCQ. The overall accuracy (%dev) from nominal low, medium and high concentrations are 8.0%, 5.3%, 2.2% for AZM; 0.0%, -0.4%, -1.3% for HCQ; -6.3%, -0.4%, -2.1% for DHCQ; and -4.9%, —5.3%, -5.6% for BDCQ (Table 4).

Intra-assay precision and accuracy were calculated from 6 replicate samples of low, medium, and high concentrations analyzed on the same day on 4 unique days. The intra-day

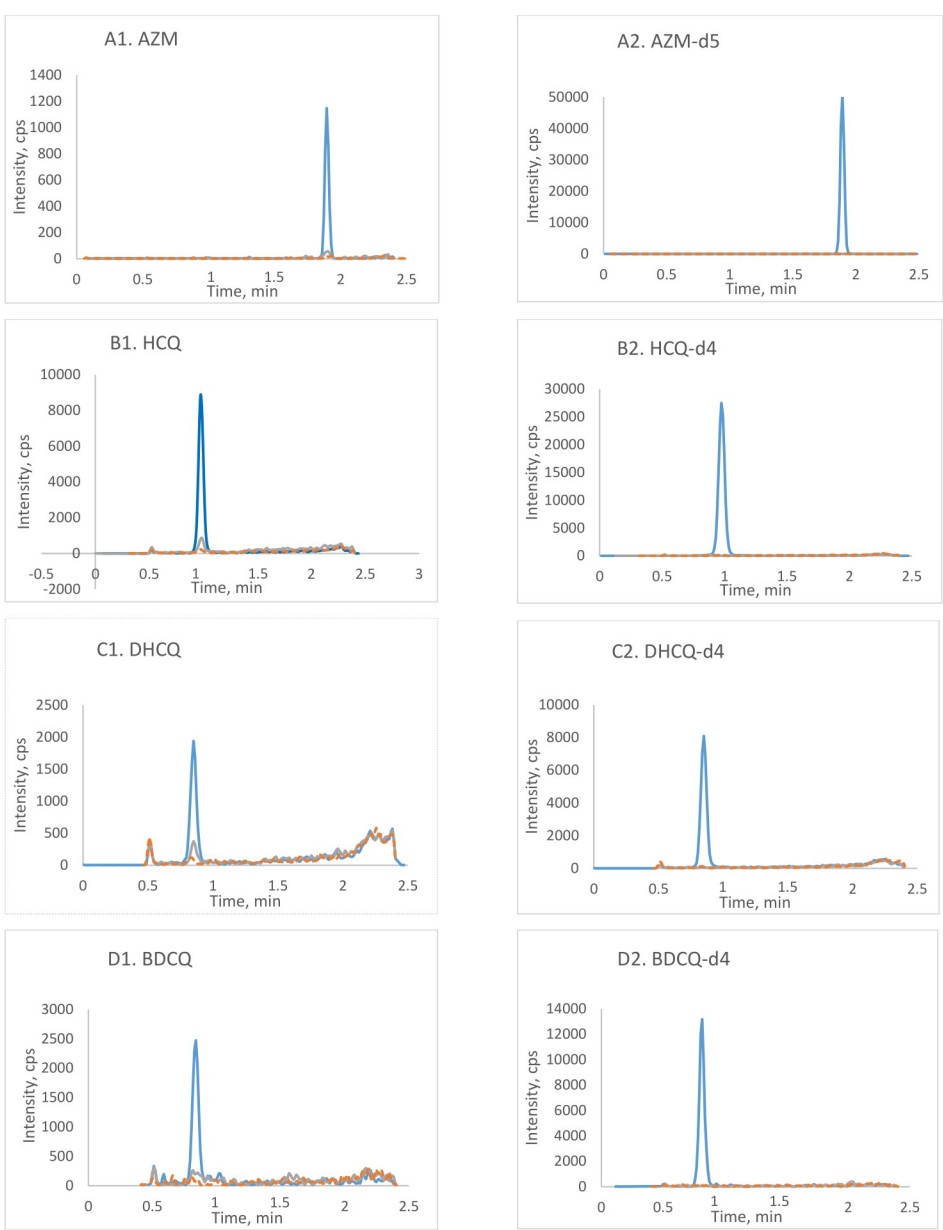

**Fig 2. Representative chromatograms of blank (red dash line), the LLOQ (blue solid line) and double blank samples following ULOQ (grey solid line).** Blank sample was processed with IS, double blank sample was processed without IS.

precision (%CV) of this method at low, medium, and high concentrations ranges from 5.5–11%, 4.1–7.2%, 2.0–4.2% for AZM; 3.6–8.9%, 3.1–6.6%, 3.5–6.7% for HCQ; 4.7–6.9%, 3.8–8.5%, 3.9–6.7% for DHCQ; and 7.6–13%, 4.7–6.0%, 4.1–12% for BDCQ. Accuracy (dev%) for low, medium and high levels ranges from 6.0–9.3%, 0.6–13%, -3.1–8.0% for AZM; -4.0–8.2%, -2.5–4.1%, -7.5–2.7% for HCQ; -13-(-0.3)%, -5.9–2.3%, -4.5–0.5% for DHCQ; and -13-8.1%, -11-2.4%, -7.8-(-3.3)% for BDCQ (Table 4).

*LLOQ.* Six replicates of validation samples at the lowest calibration concentration (2/2/1/0.5 ng/mL for AZM/HCQ/DHCQ/BDCQ) were analyzed on 4 different days to determine the inter- and intra- assay precision and accuracy of the lowest point on the calibration curve. The

**Table 4. Precision and accuracy.**

| AZM | Intra-day | | | | Inter-day | | | |
|---|---|---|---|---|---|---|---|---|
| Nominal, ng/mL | 2.00 | 6.00 | 60.0 | 800 | 2.00 | 6.00 | 60.0 | 800 |
| %RSD | 7.3–12% | 5.6–11% | 4.1–7.2% | 1.9–3.7% | 13 | 7.4 | 6.5 | 5.4 |
| %dev | -19-6.1% | 6.0–9.3% | 0.6–13% | -3.1–8.0% | -6.2 | 5.8 | 2.5 | 8.9 |
| N | 6 | 6 | 6 | 6 | 24 | 24 | 24 | 24 |
| HCQ | Intra-day | | | | Inter-day | | | |
| Nominal, ng/mL | 2.00 | 6.00 | 60.0 | 800 | 2.00 | 6.00 | 60.0 | 800 |
| %RSD | 5.7–12% | 3.6–8.9% | 3.1–6.6% | 3.5–6.7% | 12 | 7.3 | 5.3 | 6.2 |
| %dev | -3.8–19% | -4.0–8.2% | -2.5–4.1% | -7.5–2.7% | 6.4 | 0.0 | -0.4 | -1.3 |
| N | 6 | 6 | 6 | 6 | 24 | 24 | 24 | 24 |
| DHCQ | Intra-day | | | | Inter-day | | | |
| Nominal, ng/mL | 1.00 | 3.00 | 30.0 | 400 | 1.00 | 3.00 | 30.0 | 400 |
| %RSD | 9.2–13% | 4.7–6.9% | 3.8–8.5% | 3.9–6.7% | 14 | 7.5 | 6.3 | 5.1 |
| %dev | -12-13% | -13-(0.3%) | -5.9–2.3 | -4.5–0.5% | 0.8 | -6.3 | -0.4 | -2.1 |
| N | 6 | 6 | 6 | 6 | 24 | 24 | 24 | 24 |
| BDCQ | Intra-day | | | | Inter-day | | | |
| Nominal, ng/mL | 0.500 | 1.50 | 15.0 | 200 | 0.500 | 1.50 | 15.0 | 200 |
| %RSD | 9.0–14% | 7.6–13% | 4.7–6.0% | 4.1–12% | 15 | 13 | 7.4 | 8.0 |
| %dev | -11-12% | -13-8.1% | -112.4% | -7.8-(3.3)% | 0.7 | -4.9 | -5.3 | -5.6 |
| N | 6 | 6 | 6 | 6 | 24 | 24 | 24 | 24 |

inter-assay precision (%CV) is 14% for AZM, 12% for HCQ, 14% for DHCQ, and 15% for BDCQ. The inter- assay percent deviation is -7.9% for AZM, 6.4% for HCQ, 0.8% for DHCQ, and 0.7% for BDCQ. The intra-assay %CV for the mean of these 4 replicate days ranges from 7.3–12% for AZM, 5.7–12% for HCQ, 9.2–13% for DHCQ, and 9.0–14% for BDCQ. The mean accuracy (%dev) ranges from -19-6.1% for AZM, -3.8–19% for HCQ, -12-13% for DHCQ, and -11-12% for BDCQ (Table 4).

**3.2.3. Dilution integrity.** An extra-high QC plasma samples at a nominal concentration of 4000 ng/mL for AZM and HCQ, 2000ng/mL for DHCQ, and 1000ng/mL for BDCQ (4 times the ULOQ concentration) were diluted with blank plasma by 10-fold. Five replicates of the diluted samples processed and analyzed. The mean values (n = 5) were within 15% of the nominal concentration for all analytes. The %CV was 5.3%, 9.4%, 5.5% and 3.9%; and %dev was 14%, -7.0%, -11% and -10% for AZM, HCQ, DHCQ, and BDCQ, respectively, suggesting the samples can be diluted by up to 10-fold without compromising sample integrity.

**3.2.4. Stability.** *Freeze/thaw stability.* QC-L and QC-H samples undergone 4 freeze-thaw cycles were processed and analyzed along with freshly spiked calibrators and QCs. The percent differences from freshly made controls at QC-L and QC-H concentrations are -3.7% and -10% for AZM, -8.8% and -1.1% for HCQ, -12% and -5.3% for DHCQ, -7.1% and -4.2% for BDCQ (S2 Table). The percent remaining analytes compared to nominal concentrations are all within 100 (±15) % (Table 5). The results reveal that AZM, HCQ, DHCQ and BDCQ in plasma are stable after 4 freeze-thaw cycles.

*Room temperature stability in plasma.* After standing on the bench for 66 hours, *QC*-L and QC-H samples were processed and analyzed along with freshly prepared QC-L and QC-H samples and the freshly prepared calibrators. The % change from controls at QC-L and QC-H concentrations was 4.3% and 6.1% for AZM, -2.3% and 0.2% for HCQ, -7.6 and -6.8% for DHCQ, -12% and -6.3% for BDCQ, all of which were within ±15% (S2 Table). When compared to spiked nominal concentrations, the remaining concentration are all within 100 (±15)

Table 5. Stability of AZM, HCQ, DHCQ, and BDCQ (n = 3).

| Treated conditions | AZM | | HCQ | | DHCQ | | BDCQ | |
|---|---|---|---|---|---|---|---|---|
| | conc., ng/mL | % remaining | conc., ng/mL | % remaining | conc., ng/mL | % remaining | conc., ng/mL | % remaining |
| 38 days, -70˚C | | | | | | | | |
| Low | 6.13±0.26 | 102 | 5.20±0.04 | 86.7 | 2.58±0.03 | 86.0 | 1.32±0.13 | 88.0 |
| High | 859±33 | 107 | 790±32 | 98.8 | 367±25 | 91.8 | 178±16 | 89.0 |
| 66hr, 22±3˚C (RT) | | | | | | | | |
| Low | 5.98±0.13 | 99.7 | 5.46±0.32 | 91.0 | 2.74±0.12 | 91.3 | 1.30±0.07 | 86.7 |
| high | 801±33 | 100 | 786±36 | 98.3 | 363±15 | 90.8 | 177±7 | 88.5 |
| Reinjection, 3 days | | | | | | | | |
| Low | 5.67±0.35 | 94.4 | 5.81±0.13 | 96.9 | 3.16±0.08 | 105 | 1.50±0.14 | 100 |
| high | 845±31 | 106 | 718±64 | 89.8 | 410±2 | 102 | 187±12 | 93.5 |
| 4-freeze-thaw cycles | | | | | | | | |
| low | 5.14±0.36 | 85.7 | 5.78±0.02 | 96.3 | 2.92±0.12 | 97.3 | 1.49±0.12 | 99.3 |
| high | 765±23 | 95.6 | 784±17 | 98.0 | 373±11 | 93.3 | 189±2 | 94.5 |
| Blood, 6hr | | | | | | | | |
| high | 878±34 | 104 | 687±21 | 112 | 303±10 | 106 | 135±11 | 103 |
| Combined working solution, 50%MeOH | | | | | | | | |
| RT, 8 days | | 96.8 | | 96.6 | | 97.1 | | 94.5 |

*% remaining was calculated by comparing to the nominal values (100%) for plasma and reinjection stability, comparing to the spiked blood samples at 22min for blood stability, and comparing to the same solution frozen at -70˚C for solution room temperature (RT) stability.

%. The results suggest that plasma samples are stable for at least 66 hours at room temperature (Table 5).

*Reinjection reproducibility/autosampler stability*. To test autosampler stability, the analyzed samples were left in the autosampler and re-injected 3 days after (71hr). The percent remaining drug concentrations from nominal values at QC-L and QC-H were 94.4 and 106% for AZM, 96.9 and 89.8% for HCQ, 105 and 102% for DHCQ, 100 and 93.5% for BDCQ, suggesting the processed samples are stable in autosampler for at least 3 days (Table 5).

*Long-term stability at -70℃ of plasma samples*. To test long-term stability at -70˚C, triplicates of the QC-L and QC-H plasma samples stored at -70˚C for 38 days were analyzed along with freshly spiked calibrators and QC samples as controls in triplicate. The treated samples were all within ±15% difference from controls. The percent differences from control QC-L and QC-H samples are 3.7% and -3.0% for AZM, 1.3% and 2.7% for HCQ, -1.9% and -0.9% for DHCQ, and -2.0% and -5.7% for BDCQ (S2 Table). When compared to nominal values, the treated samples were also within 100 (±15) % (Table 5). The data demonstrate plasma samples are stable at -70˚C for at least 38 days. Previous study reported AZM in plasma is stable at -70˚ C for at least 92 days [26].*Stability of blood samples at room temperature*. When compared to plasma samples separated from blood at 22 min, less than 15% difference over 6 hr was found for all drugs, suggesting AZM, HCQ, DHCQ and BDCQ are stable in blood for at least 6 hr at room temperature (Table 5). When compared to nominal concentration, less than 15% change was found for AZM over 6 hr at room temperature, suggesting AZM is equally distributed in blood cells and plasma. Whereas, for HCQ and its metabolites, over 20% (-20% for HCQ, -30% for DHCQ, -35% for BDCQ) lower than nominal concentration was found at 22 min after the drugs had been spiked into blood (S2 Table), suggesting these analytes concentrated in blood cells. Previous studies reported that AZM concentrated in blood leukocytes and inflamed tissues and its concentration in blood was double of that in plasma on day 3 and

4-fold higher after day 30 [19]–Likely because AZM is slowly distributed into blood cells. HCQ level is also higher in blood than in plasma [36], consistent with our results.

*Solution stability.* AZM stock solution (2 mg/mL in 50% MeOH) was stable for at least 23 hr at room temperature and for at least 45 days at -70˚C, the % difference from the untreated fresh controls are -1.3% and -0.1% respectively. HCQ stock (2 mg/mL in water) was stable at -70˚C for at least 63 days and at room temperature (19–22˚C) for at least 5 days, with the %difference from fresh controls at 6.6% and 1.5%, respectively (S2 Table). The combined working solution (40/40/20/10 μg/mL AZM/HCQ/DHCQ/BDCQ in 50% MeOH) was stable at room temperature for at least 8 days (Table 5). Previous studies reported that AZM stock in methanol is stable for 92 days at -10˚C [26]. HCQ and DHCQ stocks (0.2 mg/mL) in water are stable at -80˚C for 12 months [18].

The working solutions for all deuterated ISs were stable at room temperature for at least 23 hr in plastic tube (S1 Table). However, due to adsorption on glass surface, the IS solution should be prepared in plastic container. Similarly, the stock and working solutions of analytes —especially those of HCQ, DHCQ and BDCQ should be prepared and stored in plastic containers—even though the impact of adsorption on glass surface may diminish at higher drug concentration. For example, AZM stock solution at 0.5 mg/mL in glass vial is comparable to those in plastic vial (<5% difference) (S2 Table).

**3.2.5. Matrix effect, recovery, and process efficiency.** Three sets of samples each at three concentration levels (low, medium, and high validation concentration levels) were prepared and analyzed to determine ME, RE, and PE. The mean peak area and peak area ratio (analyte/ IS) were calculated for each level in each set of samples and comparisons are presented in Table 6.

$$ME = \frac{100 \times \text{peak area of post extraction spiked sample (set2)}}{\text{peak area of clean sample (set1)}} \qquad (4)$$

$$RE = \frac{100 \times \text{peak area of pre extraction spiked sample (set3)}}{\text{peak area of post extraction spiked sample (set2)}} \qquad (5)$$

$$PE = \frac{100 \times \text{peak area of pre extraction spiked sample (set3)}}{\text{peak area of clean sample (set1)}} \qquad (6)$$

**Table 6. Matrix effect, recovery and process efficiency.**

| Analytes | Conc. | Matrix Effect | | Recovery | | PE | |
|---|---|---|---|---|---|---|---|
| | (ng/ml) | analyte | IS | analyte | IS | analyte | IS |
| | Low (6) | 100 | 102 | 102 | 101 | 102 | 103 |
| AZM | Med (60) | 98.8 | 96.4 | 99.4 | 100 | 98.3 | 96.8 |
| | High (800) | 97.9 | 95.7 | 98.6 | 102 | 96.5 | 97.8 |
| | Low (6) | 110 | 106 | 90.5 | 96.1 | 99.8 | 101 |
| HCQ | Med (60) | 98.0 | 101 | 94.4 | 97.2 | 92.4 | 97.7 |
| | High (800) | 103 | 100 | 88.9 | 94.1 | 91.9 | 94.1 |
| | Low (3) | 102 | 103 | 88.6 | 99.8 | 90.4 | 103 |
| DHCQ | Med (30) | 101 | 98.9 | 90.3 | 99.3 | 91.4 | 98.2 |
| | High (400) | 106 | 101 | 92.9 | 96.7 | 98.0 | 97.8 |
| | Low (1.5) | 97.1 | 104 | 90.9 | 91.8 | 88.3 | 95.6 |
| BDCQ | Med (15) | 103 | 110 | 88.7 | 94.0 | 91.6 | 104 |
| | High (200) | 101 | 101 | 89.0 | 87.7 | 90.2 | 88.4 |

*Recovery (RE).* The recovery of analytes from plasma following sample preparation was assessed by comparing the peak areas from set 3 and set 2. The recoveries for AZM were 102, 99.4 and 98.6% at low, medium, and high concentration, respectively, and the recovery for the IS ranged from 100–102%. The recoveries ranged from 88.9–94.4% for HCQ, 88.6–92.9% for DHCQ, and 88.7–90.9% for BDCQ. The %CV of peak areas for recovery experiment was all within 12%. These results suggest the assay is highly reproducible across the concentration range with consistent and high recovery.

*Matrix Effect (ME).* The matrix effect of analytes from plasma following sample preparation was assessed by comparing the peak areas from set 2 and set 1. The ME for AZM were 100, 98.8 and 97.9% at low, medium, and high concentration, respectively. ME for the IS ranged from 95.7–102%. The ME for HCQ, DHCQ and BDCQ ranged from 98.0–110%, 101–106%, 97.1–103%, respectively. The IS normalized ME were within (100±10) % for all 4 analytes.

*Process Efficiency (PE).* The PE of analytes from plasma following sample preparation was assessed by comparing the peak areas from set 3 and set 1. The PE for AZM were 102, 98.3 and 96.5% at low, medium, and high concentration, respectively. ME for the IS ranged from 96.8–103%. The PE for HCQ, DHCQ and BDCQ ranged from 91.9–99.8%, 90.4–98.0%, 88.3–91.6%, respectively.

**3.2.6. Selectivity.**   To test selectivity, 6 lots of blank plasma were processed without adding ISs and analyzed along with a LLOQ sample. The results are shown in Fig 3. significant signals were found at the retention times of both analyte and IS for each analyte. The data suggest the method is highly selective.

**3.2.7. Impact of hemolysis.**   Compared to controls, the differences of hemolyzed plasma samples at QC-L and QC-H concentrations were within 15% for all analytes (10.8 and -1.4% for AZM, 3.0 and 0.94% for HCQ, 6.3% and -8.1% for DHCQ, -4.3 and -3.0 for BDCQ) (S2 Table). The results suggest hemolysis does not impact quantitation of the analytes in this assay. However, since AZM and HCQ are known to present higher in blood than plasma, it is recommended to avoid hemolysis when processing clinical samples.

**3.2.8. Impact of anticoagulant counter ions.**   The differences of $K_2$EDTA plasma samples from $K_3$EDTA plasma samples (controls) were all within ±15% at QC-L and QC-H concentrations. For AZM, the differences from the controls at QC-L and QC-H were -1.6% and-8.0%, -4.2% and -5.0% in the two lots of $K_2$EDTA plasma. For HCQ, the differences at QC-L and QC-H were 4.6% and -6.3%, 7.3% and 2.1% in the two lots of $K_2$EDTA plasma. For DHCQ, they are -3.9% and -13%, 9.1% and 0.1%. For BDCQ, they are-3.1% and -11%, 11% and -0.8%, respectively (S2 Table). The data suggest different counter ions in anticoagulant won't interfere assay performance, and $K_2$EDTA collection tubes could be used as an alternative for sample collection.

## 4. Conclusions

A high-throughput method for simultaneous quantitation of HCQ, DHCQ, BDCQ and AZM in plasma was developed and validated based on guidelines from FDA and NIH-sponsored CPQA, and suitable for clinical studies of those drugs. The method required only 20 µL plasma sample and 3.5 min run time and carryover is negligible. Although clinical data for HCQ and AZM use in COVID-19 have resulted in poor outcomes, understanding the pharmacological basis for toxicity is still of interest to inform any potential future use of these compounds. Therefore, methods to quantitate these drugs reliably still remain relevant. Furthermore, these compounds continue to be used or evaluated for other illnesses besides COVID-19 such as malaria, inflammatory diseases, and other viral and bacterial infections [37].Such evaluations will require state of the art analytical methods such as one described here. For it requires only

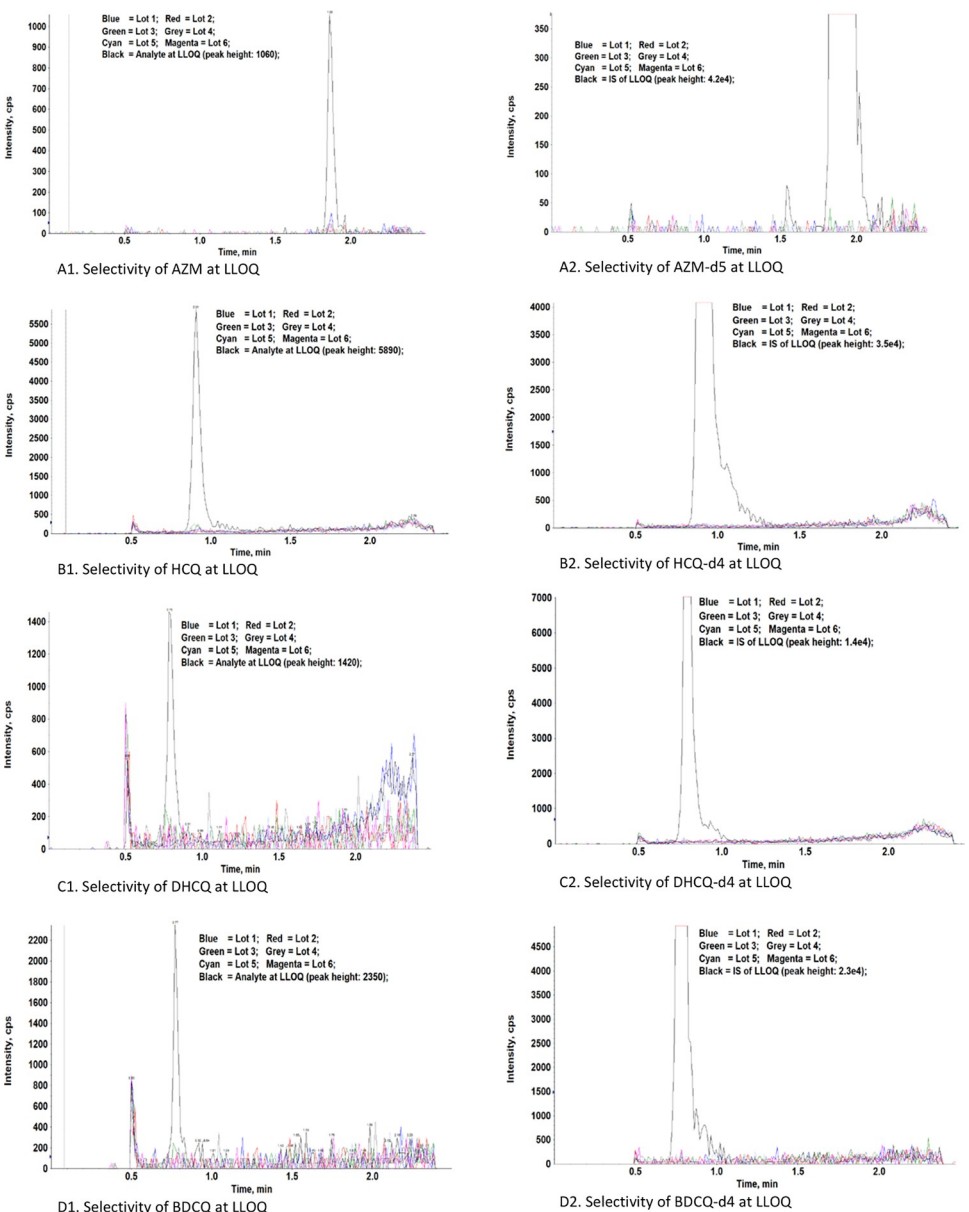

**Fig 3. Chromatograms of six lots of blank plasma and LLOQ sample.**

a small sample volume, our method can be used for pediatric studies where sample volume is limited. and it can be coupled with capillary tube sampling by a finger prick or more advanced microsampling techniques such as Seventh Sense Tap™ to facilitate clinical studies. With the highly sensitive LC-MS/MS system, our method may also be modified for dried blood spot samples.

## Supporting information

**S1 Fig. Linearity of AZM calibration curve in ESI versus APCI.**
(PDF)

**S2 Fig. LLOQ versus ULOQ chromatograms.**
(PDF)

**S1 Table. Adsorption of internal standards on container surface.**
(DOCX)

**S2 Table. Stability data.**
(PDF)

**S1 File. Assay standard operating procedure.**
(PDF)

## Acknowledgments

We wish to thank supporting staff at University of California San Francisco (UCSF). We also want to thank Difrancesco Robin and Andrew Ocque from University of New York at Buffalo, and Lane Bushman from University of Corolado for their constructive discussion on the method validation and thank supporting staff and reviewers for CPQA.

## Author Contributions

**Conceptualization:** Francesca Aweeka.

**Data curation:** Vong Sok, Florence Marzan, David Gingrich, Liusheng Huang.

**Formal analysis:** Liusheng Huang.

**Funding acquisition:** Francesca Aweeka.

**Methodology:** Liusheng Huang.

**Project administration:** Florence Marzan.

**Resources:** Florence Marzan, David Gingrich.

**Supervision:** Francesca Aweeka.

**Validation:** Vong Sok, Liusheng Huang.

**Writing – original draft:** Vong Sok, David Gingrich, Liusheng Huang.

**Writing – review & editing:** Vong Sok, Florence Marzan, David Gingrich, Francesca Aweeka, Liusheng Huang.

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
