## [Decision Letter · Decision Letter 0]

23 Dec 2020

PONE-D-20-26989

A validated LC-MS/MS method for simultaneous determination of azithromycin, hydroxychloroquine, and its metabolites desethylhydroxychloroquine and bisdesethylchoroquine in human plasma

PLOS ONE

Dear Dr. Huang,

Thank you for submitting your manuscript to PLOS ONE. After careful consideration, we feel that it has merit but does not fully meet PLOS ONE’s publication criteria as it currently stands. Therefore, we invite you to submit a revised version of the manuscript that addresses the points raised during the review process.

We look forward to receiving your revised manuscript.

Kind regards,

Pasquale Avino, Ph.D.

Academic Editor

PLOS ONE

2. For reproducibility purposes please clearly specify in your methods section the source of the blood and plasma used in your study (eg. brand, product number)

"This work was partially supported by the National Institutes of Health (NIH) through AIDS clinical Trials Group (ACTG), grant number 1UM1 AI068636 (F.A.). URL: www.NIH.gov. The funder plays no role in the study or prepataion of the manuscript."

Reviewers' comments:

Reviewer #1: Comments to Author:

A validated LC-MS/MS method for simultaneous determination of azithromycin, hydroxychloroquine, and its metabolites desethylhydroxychloroquine and

bisdesethylchoroquine in human plasma

The paper presented in this study is a validated LC-MS/MS method for simultaneous determination of Hydroxychloroquine (HCQ) and azithromycin (AZM) are antimalarial drugs recently reported to be active against severe acute respiratory syndrome coronavirus-2 (SARS-CoV-2) and quantitation method to assess the pharmacokinetics of AZM, HCQ, and its metabolites desethylhydroxychloroquine (DHCQ) and bisdesethylchloroquine (BDCQ) in patients’ plasma. The work is well written and organized and the results are satisfactorily supported by the reported data. Figures and tables are comprehensive and helpful. However, some minor changes are required as discussed in the following:

1. Abstract should be limited to 200 words

2. Keywords should not be the repetitions of the title words, please find such words which are not in the title, this way search engines of the web will find your manuscript with higher probability.

3. Suggestions for improvements in the Title

4. The structure of scientific publication should include the general chapters (Introduction, Materials and Methods, Results, Discussion, Conclusion). Please follow instructions on journal webpage. Conclusion chapter is missing.

5. Recommendations for future studies are needed in the conclusion section. Kindly provide strong recommendations for future researches.

6. Some citations are missing from the References section.

7. Some references are not cited in the text.

8. Formatting does not match journal criteria. (e.g. References section; section title spacing; paragraph indents)

The paper generates the following kinds of data.

1. Only 20 µL plasma sample by volume is needed for simultaneous quantitation of AZM, HCQ, DHCQ, and BDCQ.

2. The run time is 3.5 min which is fast turnaround time.

3. Statistical measurements are missing in whole of the data which should be added in the manuscript.

However, before I can recommend its publication, the authors should address the following questions

Some questions to author

1. The %differences from controls are 3.8% and -2.9% for AZM, 1.3% and 2.7% for HCQ, -1.9% and -0.9% for DHCQ, and -2.0% and -5.7% for BDCQ. Can author explain what would be the structural changes?

2. Why the study is important, what benefits readers can get from this study?

Reviewer #2: This is an interesting manuscript describing a method to analyze the compounds of interest. The authors follow the FDA and CPQA methodology. The quality of the data and results are outstanding. There are several minor points that the authors should address prior to acceptance of the manuscript. These points are listed below in order of appearance in the paper.

1. Page 5, line 3: There are two periods after [21].

2. Page 6, Figure 1: I recommend having the same relative geometry of the rings in the structure for the last 6 compounds to aid in visualizing the differences between structures.

3. Page 6, Section 2.2: PFP should be defined for the more general reader.

4. Page 7, section 2.4: HLB should be defined for the more general reader.

5. Page 5 and 22-23: These equations should be reformatted and numbered using the recommended standards of the journal.

6. Page 19, end of Room Temperature Stability in Plasma section: (Table] should be replaced with (Table 5).

---

## [Author Response · Author response to Decision Letter 0]

16 Jan 2021

Comments from the Editor’s office.

Response: The manuscript was revised with updated format and style. 

2. For reproducibility purposes please clearly specify in your methods section the source of the blood and plasma used in your study (eg. brand, product number)

 Response: blank blood and plasma sources are now added.

"This work was partially supported by the National Institutes of Health (NIH) through AIDS clinical Trials Group (ACTG), grant number 1UM1 AI068636 (F.A.). URL: www.NIH.gov. The funder plays no role in the study or prepataion of the manuscript."

 Response: amended funding statement was included in this letter.

 Response: We addressed the “data not shown” issues by providing the data in Supporting S2 Figure and S2 Tables.

Reviewers' comments:

Reviewer #1: Comments to Author:

1. Abstract should be limited to 200 words

Response: The abstract was truncated but still over 300 words limit. We deem exception should be granted in this case as this is a comprehensive assay with 4 analytes.

2. Keywords should not be the repetitions of the title words, please find such words which are not in the title, this way search engines of the web will find your manuscript with higher probability.

Response: additional key words are included: malaria; solid-phase extraction;

3. Suggestions for improvements in the Title

Response: The title is updated as follows: Development and validation of an LC-MS/MS method for determination of hydroxychloroquine, its two metabolites, and azithromycin in EDTA-treated human plasma.

4. The structure of scientific publication should include the general chapters (Introduction, Materials and Methods, Results, Discussion, Conclusion). Please follow instructions on journal webpage. Conclusion chapter is missing.

Response: we have revised the conclusion section (4), and discussion was incorporated in the results section. 

5. Recommendations for future studies are needed in the conclusion section. Kindly provide strong recommendations for future researches.

Response: In Conclusion section, we discussed the potential use of the method for studies other than Covid-19. We also added the following sentences: 

“For it requires only a small sample volume, our method can be used for pediatric studies where sample volume is limited, and it can be coupled with capillary tube sampling by a finger prick or more advanced microsampling techniques such as Seventh Sense Tap� to facilitate clinical studies. With the highly sensitive LC-MS/MS system, our method may be modified for dried blood spot samples.”

6. Some citations are missing from the References section.

Response: The citations in text can all be found in references section

7. Some references are not cited in the text.

Response: All 37 references are now cited in the text using ENDNote program.

8. Formatting does not match journal criteria. (e.g. References section; section title spacing; paragraph indents)

The paper generates the following kinds of data.

1. Only 20 µL plasma sample by volume is needed for simultaneous quantitation of AZM, HCQ, DHCQ, and BDCQ.

2. The run time is 3.5 min which is fast turnaround time.

3. Statistical measurements are missing in whole of the data which should be added in the manuscript.

However, before I can recommend its publication, the authors should address the following questions

Some questions to author

1. The %differences from controls are 3.8% and -2.9% for AZM, 1.3% and 2.7% for HCQ, -1.9% and -0.9% for DHCQ, and -2.0% and -5.7% for BDCQ. Can author explain what would be the structural changes?

2. Why the study is important, what benefits readers can get from this study?

Response: Formatting was updated. The % differences were calculated from the measured control QC-L and QC-H concentrations. The data to calculate the % differences are now provided in Supporting S2 tables.

The method was developed to support potential clinical studies. Readers in the field of COVID-19, malaria, and inflammatory diseases using these drugs may be interested in the method and drug stability data. The small sample volume used in our method will benefit pediatric studies as stated in the conclusion section now. Furthermore, the approaches for method development are exemplary for readers in bioanalysis field,. 

Reviewer #2: 

1. Page 5, line 3: There are two periods after [21].

Response: The extra period is removed now.

2. Page 6, Figure 1: I recommend having the same relative geometry of the rings in the structure for the last 6 compounds to aid in visualizing the differences between structures.

Response: The Figure 1 is updated.

3. Page 6, Section 2.2: PFP should be defined for the more general reader.

Response: It is now defined as follows: pentafluorophenyl (PFP)

4. Page 7, section 2.4: HLB should be defined for the more general reader.

Response: It is now defined as follows: Hydrophilic lipophilic balance (HLB) solid phase extraction.

5. Page 5 and 22-23: These equations should be reformatted and numbered using the recommended standards of the journal.

Response: The equations are reformatted and numbered now.

6. Page 19, end of Room Temperature Stability in Plasma section: (Table] should be replaced with (Table 5). 

Response: It is now corrected.

---

## [Editor Report · Decision Letter 1]

8 Feb 2021

Development and validation of an LC-MS/MS method for determination of hydroxychloroquine, its two metabolites, and azithromycin in EDTA-treated human plasma

PONE-D-20-26989R1

Dear Dr. Huang,

We’re pleased to inform you that your manuscript has been judged scientifically suitable for publication and will be formally accepted for publication once it meets all outstanding technical requirements.

Kind regards,

Pasquale Avino, Ph.D.

Academic Editor

PLOS ONE

---

## [Editor Report · Acceptance letter]

19 Feb 2021

PONE-D-20-26989R1 

Development and validation of an LC-MS/MS method for determination of hydroxychloroquine, its two metabolites, and azithromycin in EDTA-treated human plasma 

Dear Dr. Huang:

I'm pleased to inform you that your manuscript has been deemed suitable for publication in PLOS ONE. Congratulations! Your manuscript is now with our production department. 

Kind regards, 

on behalf of

Professor Pasquale Avino 

Academic Editor

PLOS ONE